# Increased H⁺ efflux is sufficient to induce dysplasia and necessary for viability with oncogene expression

**Bree K Grillo-Hill, Changhoon Choi, Maite Jimenez-Vidal, Diane L Barber\***

Department of Cell and Tissue Biology, University of California, San Francisco, San Francisco, United States

**Abstract** Intracellular pH (pHi) dynamics is increasingly recognized as an important regulator of a range of normal and pathological cell behaviors. Notably, increased pHi is now acknowledged as a conserved characteristic of cancers and in cell models is confirmed to increase proliferation and migration as well as limit apoptosis. However, the significance of increased pHi for cancer in vivo remains unresolved. Using *Drosophila melanogaster*, we show that increased pHi is sufficient to induce dysplasia in the absence of other transforming cues and potentiates growth and invasion with oncogenic Ras. Using a genetically encoded biosensor we also confirm increased pHi in situ. Moreover, in *Drosophila* models and clonal human mammary cells we show that limiting H⁺ efflux with oncogenic Raf or Ras induces acidosis and synthetic lethality. Further, we show lethality in invasive primary tumor cell lines with inhibiting H⁺ efflux. Synthetic lethality with reduced H⁺ efflux and activated oncogene expression could be exploited therapeutically to restrain cancer progression while limiting off-target effects.

**\*For correspondence:** diane. barber@ucsf.edu

**Competing interests:** The authors declare that no competing interests exist.

## Introduction

Dysregulated pH is a common characteristic of cancer cells, which have a lower extracellular pH (pHe) and higher intracellular pH (pHi) than normal cells. The lower pHe of tumors, confirmed by pH-sensitive PET radiotracers, MR spectroscopy and MRI (*Zhang et al., 2010*), contributes to local metastatic invasion (*Cardone et al., 2005*; *Rofstad et al., 2006*; *Stock et al., 2008*; *Stock and Schwab, 2009*; *Busco et al., 2010*; *Estrella et al., 2013*). Increased pHi enables a number of cancer cell behaviors, including promoting cell proliferation (*Pouysségur et al., 1982*; *Kapus et al., 1994*; *Putney and Barber, 2003*), glycolytic metabolism (*Reshkin et al., 2000*; *Dietl et al., 2010*; *Webb et al., 2011*), migration (*Denker and Barber, 2002*; *Patel and Barber, 2005*; *Frantz et al., 2008*; *Stock and Schwab, 2009*), and invasion (*Reshkin et al., 2000*; *Hinton et al., 2009*), as well as limiting apoptosis (*Matsuyama et al., 2000*; *Lagadic-Gossmann et al., 2004*). The higher pHi of cancer cells is paradoxical because increased production of metabolic acids generated by aerobic glycolysis would be predicted to lower pHi. However, many cancers have elevated expression or activity of proteins that facilitate increased pHi, including carbonic anhydrase 9 (*Swietach et al., 2007*), H⁺-ATPases (*Martinez-Zaguilan et al., 1993*; *Sennoune et al., 2004*; *Hinton et al., 2009*), the ubiquitously expressed Na⁺-H⁺ exchanger NHE1 (*McLean et al., 2000*; *Miraglia et al., 2005*; *Chiang et al., 2008*; *Yang et al., 2011*) and the monocarboxylate transporter family members MCT1 and MCT4 (*Pinheiro et al., 2010*; *Halestrap, 2013*). Therapeutic targeting of these proteins to reduce H⁺ efflux and lower pHi has been suggested for limiting cancer progression (*Webb et al., 2011*; *Harguindey et al., 2013*), based primarily on findings with xenograft models and isolated cells. Xenograft tumor growth is suppressed by inhibiting NHE1 (*Lagarde et al., 1988*; *Yang et al., 2011*) or MCT1 (*Sonveaux et al., 2008*; *Colen et al., 2011*) activity. In cell models, inhibiting NHE1 activity reduces viability of breast cancer (*Reshkin et al., 2003*) and leukemic cells (*Rich et al., 2000*; *Reshkin et al., 2003*), and silencing MCT4 expression with RNA

**eLife digest** An individual can develop cancer if cells in their body gain genetic mutations that enable the cells to divide more rapidly and move—or metastasize—to other tissues and organs. These mutations can alter the chemistry of the cell; for example, the inside of a cancer cell is much more alkaline (has a higher pH) than the inside of a normal cell. This helps the cancer cells to grow and divide rapidly, and move to other parts of the body, but it is not clear how important this change in pH within the cell is for the development of cancer.

Previous studies have shown that many cancer cells increase the amounts or activities of the proteins that remove hydrogen ions from the cell, and so make the inside of a cell more basic. Here, Grillo-Hill et al. increased the amount of one of these proteins, NHE1, in fruit flies to study its effects on healthy cells.

The experiments showed that the larval cells that produced more NHE1 were more alkaline than normal cells, and this caused the cells to become less organized and grow more rapidly. Grillo-Hill et al. also found that NHE1 can work together with the oncogene RasV12 that promotes cancer development to enhance migration within the developing wings of the larvae.

Next, Grillo-Hill et al. tested whether the change in pH is necessary for cells to become cancerous. Flies missing the gene that makes NHE1 appeared normal. However, if these cells also have the oncogene Raf—which, like RasV12, promotes cancer—the inside of the cells gradually became so acidic that they died. Grillo-Hill et al. also found that blocking the activity of NHE1 in human cancer cells could lead to increased death but does not increase the death of normal cells.

Grillo-Hill et al.'s findings show that increasing the internal pH of healthy cells leads to characteristics similar to those found in cancer cells. Furthermore, cancer cells can die if they become more acidic. This suggests that targeting pH levels could guide the development of treatments for cancer that selectively kill cancer cells while leaving normal cells unharmed.

interference decreases survival of renal carcinoma cells (*Gerlinger et al., 2012*). Although previous studies suggest important roles for pH dynamics in regulating cancer cell behaviors, whether increased pHi is sufficient or necessary for cancer progression in vivo remains unresolved.

To better understand how $H^+$ efflux and increased pHi affect cancer cell behaviors we asked two distinct but related questions. First, is increased pHi by NHE1 over-expression in the absence of other transforming signals sufficient to induce dysplasia? Second, is $H^+$ efflux by NHE1 necessary for oncogene-induced dysplasia? We found that over-expression of *Drosophila melanogaster Dnhe2*, an ortholog of mammalian NHE1, causes dysplasia, increases proliferation and facilitates oncogene-induced cell invasion in vivo. Additionally, we found that reducing $H^+$ efflux genetically or pharmacologically limits oncogene-induced increases in proliferation and has synthetic lethality with oncogenic Raf in the *Drosophila* retina as well as in human mammary epithelial cells expressing oncogenic RasV12.

## Results

To address our first question we overexpressed *Dnhe2* in *Drosophila* eye imaginal discs. Three NHE isoforms have been identified in *Drosophila (Dnhe1-3)*, and sequence analysis of the C-terminal cytoplasmic domains suggests *Dm*Nhe2 is most homologous to the mammalian plasma membrane NHE1, one of nine mammalian isoforms (*Figure 1A*). Conserved features include a lysine/arginine-rich region (KR motif) in the juxtamembrane segment that binds phosphatidylinositol 4,5-bisphosphate in the plasma membrane (*Putney and Barber, 2003*), a conserved binding motif for the calcineurin homologous protein CHP (*Lin and Barber, 1996*; *Pang et al., 2001*), a glutamic acid residue in the transmembrane domain that is essential for $H^+$ efflux (E358 in *Dm*Nhe2 and E266 in human NHE1) and consensus sites for phosphorylation by Akt and ATM/ATR kinases (*Figure 1A*, and *Figure 1—figure supplements 1, 2*). The only previous report on *Dm*Nhe2 function indicates a role in charge-dependent plasma membrane recruitment of Dishevelled to establish planar cell polarity (*Simons et al., 2009*).

We used the *GMR-GAL4* driver to over-express *Dnhe2* (*GMR > Dnhe2*) in wandering third larval instar (wL3) eye imaginal discs, a developmental stage with proliferating cells that marks the onset of retinal pattern formation. *GMR > Dnhe2* expression induced an externally rough adult eye with

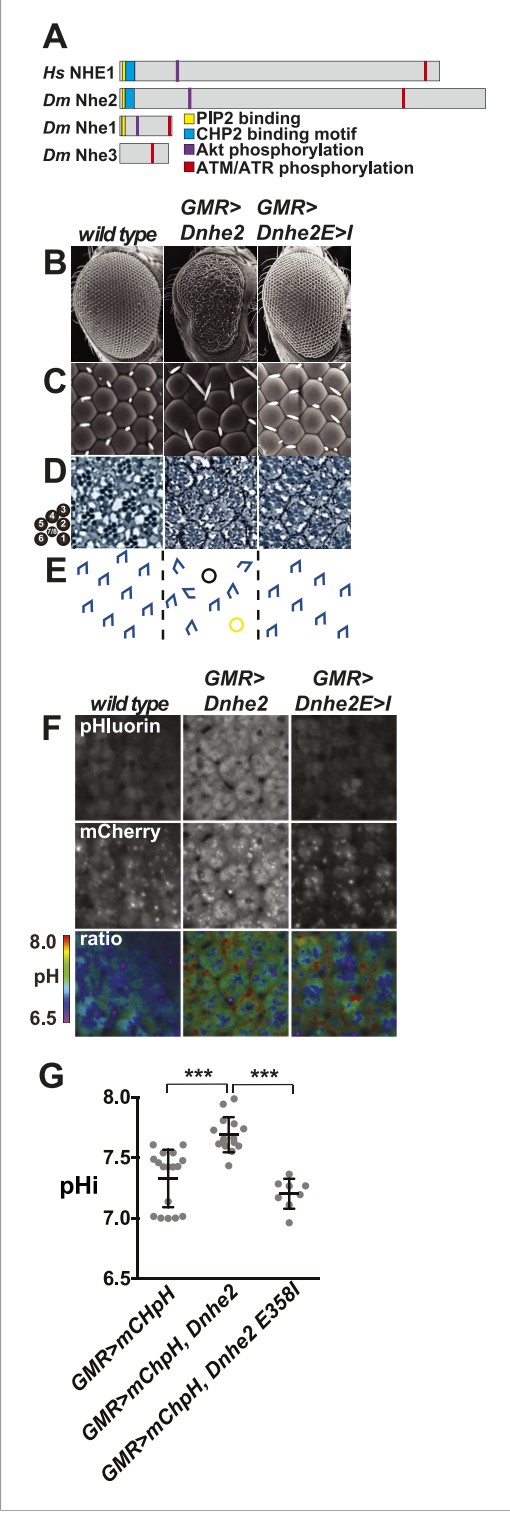

**Figure 1**. *Dnhe2* over-expression is sufficient to induce dysplasia and hyperproliferation. (**A**) Amino acid sequence analysis of NHE C-terminal cytoplasmic domains reveals that *Drosophila melanogaster* (*Dm*) Nhe2 has the highest homology with mammalian NHE1 compared with *Dm*Nhe1 and 3. (**B** and **C**) Scanning electron micrographs of the exterior surface of adult Drosophila

phenotypes consistent with dysplasia in the underlying epithelium (*Figure 1B*). In wild type *Drosophila*, the adult eye is a precisely patterned epithelial structure with hexagonal unit eyes, termed ommatidia, uniformly arrayed across the surface (*Figure 1B,C*, left). Wild type ommatidia contain eight photoreceptor neurons (R cells), forming a polarized, asymmetrical trapezoid (*Figure 1D*, tracing in *Figure 1E*, left). In *GMR > Dnhe2* flies, ommatidial organization was disrupted and individual facets were irregular in shape and size (*Figure 1B,C*, middle). In section, *GMR > Dnhe2* retinae had misoriented ommatidia, abnormal cell fates and missing R cells (*Figure 1D,E*, middle). This phenotype is similar to disrupted tissue organization with expression of a short, unregulated isoform of *Dnhe2* in a subset of retinal cells using *sevGAL4* (*Simons et al., 2009*). We also used *eyelessGAL4* to express *Dnhe2* earlier in eye development, which caused a very mild rough eye phenotype. To resolve whether increased H⁺ efflux by *Dnhe2* is necessary for disrupted tissue architecture we generated transgenic flies expressing a mutant *Dnhe2^{E358I}*, analogous to the NHE1^{E266I} mutation that abrogates H⁺ efflux (*Denker and Barber, 2002*). Externally, *GMR > Dnhe2^{E358I}* retinae had subtle defects in patterning, including rare bristle placement defects (*Figure 1B,C*, right). In section, no patterning defects were evident (*Figure 1D,E*, right), suggesting that dysplasia in *GMR > Dnhe2* eyes is dependent on ion transport.

To determine whether *Dnhe2* over-expression increased pHi we generated transgenic flies expressing a genetically encoded, ratiometric mCherry-pHluorin pH sensor that was previously used to measure pHi in cultured cells (*Koivusalo et al., 2010*; *Choi et al., 2013*). After ~24 hr of transgene expression there was no detectable difference in pHi between wild type and *GMR > Dnhe2* in wL3 eye imaginal discs (data not shown). However, in pupal eyes at 42 hr after puparium formation (42H apf), pHi in *GMR > Dnhe2* was significantly higher than wild type in both apical non-neural cells (*Figure 1F,G*) and in photoreceptor neurons (*Figure 1—figure supplement 3*). In contrast, pHi in pupal eyes expressing the mutant *GMR > Dnhe2^{E358I}* was not different than wild type (*Figure 1F,G*), indicating that this mutant lacks H⁺ efflux. The delay between onset of transgene expression and increased pHi could be due to accumulation of functionally processed *D*Nhe2 protein. These data suggest that patterning defects in *GMR > Dnhe2* retinae reflect constitutively

*Figure 1. Continued*

eyes of the indicated genotypes. Anterior is to the right in all figures unless otherwise indicated. (**D**) Histological sections reveal an asymmetrical trapezoidal arrangement of rhabdomeres, the light-sensing organelles of the R cells, visible as dark circles; seven of the eight rhabdomeres are visible in each section. (**E**) Traced diagram showing ommatidial orientation, where blue trapezoids indicate wild type, black circles indicate missing R cells, and yellow circles indicate abnormal symmetrical R4/R4 cell specification. (**F**) Single confocal slices from live, pupal eyes expressing *GMR > mChpH* showing fluorescence of pHluorin (top), mCherry (middle) and ratiometric images (bottom). (**G**) pHi was calculated from ratio values using a standard curve and graphed (black lines show mean ± SEM). Calculated pHi values are: wild type (7.3 ± 0.06, n = 17), *GMR > Dnhe2* (7.7 ± 0.04, n = 15); *GMR > Dnhe2E358I* (7.2 ± 0.04, n = 8). (***p < 0.001).

The following figure supplements are available for figure 1:

**Figure supplement 1**. *D*Nhe2 is the ortholog of human NHE1.

**Figure supplement 2**. Heterologously expressed *D*Nhe2 in NHE1-deficient CCL39 Chinese hamster lung fibroblasts (termed PS120 cells) localizes to plasma membrane protrusions, as has been shown for mammalian NHE1 (*Putney and Barber, 2003*).

**Figure supplement 3**. *Dnhe2* over-expression increases pHi in photoreceptor neurons.

increased pHi, although we cannot rule out possible effects of decreased pHe or altered sodium levels.

To determine the underlying cause of the adult rough eye phenotype, we examined *GMR > Dnhe2* retinae at earlier time points in development. In wild-type pupal eyes, precise retinal organization is apparent on the apical surface with immunolabeling of the adherens junction protein beta-catenin (*Figure 2A*, schematic drawing on right). Four central cone cells (blue) display stereotyped contacts, and are enwrapped by two semi-circular primary pigment cells (orange) to form the ommatidial core. Individual ommatidia are separated by a single layer of secondary pigment cells (yellow), with tertiary pigment cells (green) and bristles (purple) at alternating vertices to form the hexagonal shape of ommatidia; these cells are collectively referred to as lattice cells. *GMR > Dnhe2* pupal eyes had severely disrupted cell shapes and tissue organization, and ommatidia overall were smaller than in wild type. Ommatidia had aberrant numbers of cone and primary pigment cells (pink). Some ommatidia were fused (red), reflecting fused lenses seen in adult eyes (*Figure 1B*). Filamentous actin organization (labeled by rhodamine-conjugated phalloidin, purple, *Figure 2A*) in wild type eyes showed increased labeling in cone cells and lattice cells compared to the primary pigment cells. In *GMR > Dnhe2* pupal retinae, overall phalloidin levels were lower, and the differences between cell types were less pronounced.

Disorganization was present through the depth of the pupal retinal epithelium. At the level of cone cell nuclei (*Figure 2B*, labeled 'c'), 4 nuclei per ommatidium are present in wild type, and phalloidin labels the center of each ommatidium. The number and arrangement of nuclei were aberrant with *GMR > Dnhe2* expression, although phalloidin labeling appeared similar to wild type. Optical sections ~3 μm deeper in the epithelium reveal R cell nuclei arranged in a characteristic pattern revealed by the neuronal antigen Elav, where four R cells (R1/3/6/7) are visible in wild type. We saw strong phalloidin labeling in the center of each ommatidium, marking the nascent rhabdomeres. In *GMR > Dnhe2* pupal eyes, the number and arrangement of R cells in this section were disrupted, and we did not see the phalloidin signal in the center of the ommatidia. Optical sections ~3 μm deeper in wild type pupal eyes show the stereotyped arrangement of the other four R cells (R2/5/7/8). In *GMR > Dnhe2* retinae, this organization was disrupted, and there are more Elav-negative nuclei present than in wild type. On the basal side of the WT pupal eyes, phalloidin-labeled asters mark the centers of ommatidia, and Elav-negative nuclei of lattice cells were seen. In *GMR > Dnhe2* eyes, phalloidin-labeling was decreased and did not show the same pattern as in wild type. Additionally, Elav-positive nuclei persist in this layer, suggesting severe disorganization of R cells. There were also disorganized Elav-negative nuclei, which may be lattice cells. From these data, increasing pHi in the developing eye causes profound defects in tissue architecture that affect all cell types in the developing *Drosophila* eye, and shows severe disruption of patterning throughout this complex, multilayered epithelium. These phenotypes are consistent with dysplasia, and are similar to effects of expressing activated, oncogenic Raf, as shown below.

Consistent with increased pHi being a conserved proliferative signal from yeasts (*Orij et al., 2012*) to mammals (*Pouysségur et al., 1982*; *Kapus et al., 1994*; *Putney and Barber, 2003*), cell proliferation was substantially increased with *GMR > Dnhe2* expression. A regulated band of proliferation in wL3 eye

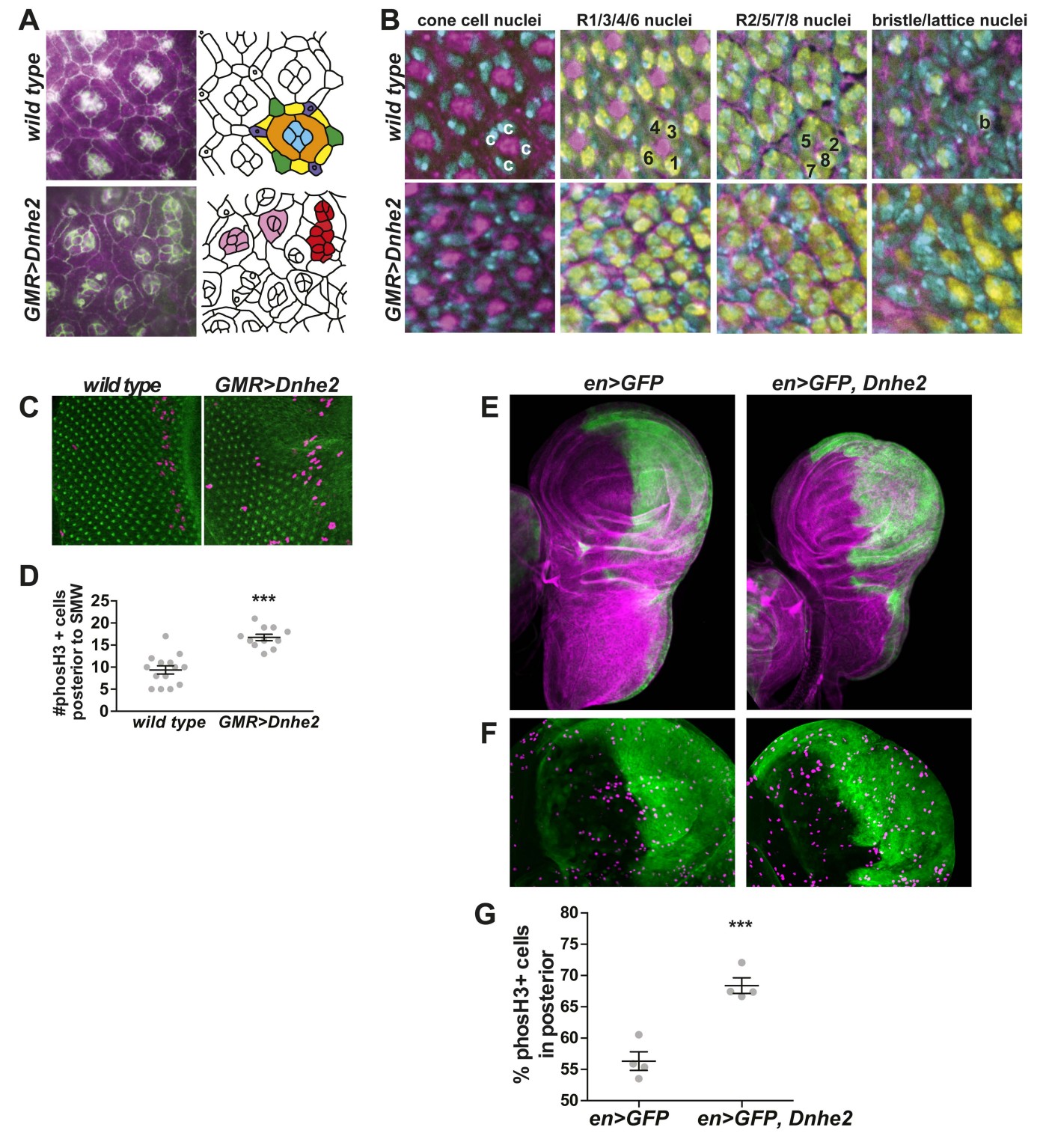

**Figure 2**. *Dnhe2* over-expression disrupts cell shape and tissue organization and increases proliferation. (**A**) Confocal micrographs of pupal retinae show disruption of cell shapes and cell–cell contacts as outlined by beta-catenin labeling (green) and rhodamine-phalloidin (magenta) in wild type (top) and *GMR > Dnhe2* pupal retinae (bottom). Schematic diagrams are shown to the right with cell types designated by colors: cone cells (blue), primary pigments cells (orange), secondary pigment cells (yellow), tertiary pigment cells (green), and bristle cells (purple). (**B**) Three-dimensional tissue architecture is shown through the depth of the epithelium. Cells are labeled as follows: DNA label Hoescht (cyan); rhodamine-phalloidin (magenta); pan neuronal marker Elav (yellow). In wild type retinae, four cone cell nuclei are seen (marked with 'c') and F-actin labeling in the center of ommatidia. 3 μm lower, the stereotyped

*Figure 2. continued on next page*

Figure 2. Continued

photoreceptor cell (R) organization where R1/6/3/4 are labeled and F-actin staining in the center of each ommatidium is seen. In the third slice, 3 µm lower, R2, 5 and 8 can be seen. And finally, on the basal side of the epithelium, the F-actin asters that mark the centers of each ommatidium and the single Elav-positive nucleus of the bristle complex found in wild type. (**C**) Immunolabeling of wL3 eye imaginal discs showing proliferating cells (phosH3, magenta) and adherens junctions (beta-catenin, green). (**D**) Quantification of proliferating cells posterior to the SMW (mean ± SEM: wild type (9.4 ± 0.9, n = 14), *GMR > Dnhe2* (16.7 ± 0.71, n = 11). (**E**) Confocal slices through whole mount wing discs with *engrailed-GAL4* (*enGAL4*) driver driving expression of GFP (green), or GFP and *Dnhe2*. Wing discs are labeled with rhodamine-phalloidin (magenta) to show tissue architecture. (**F**) An enlarged view of the wing pouch area, labeled with *en > GFP* (green) and phosH3 (purple). (**G**) Percent of phosH3+ cells in the posterior compartment (mean ± SEM: *en > GFP* [56.3 ± 1.5, n = 4]; *en > GFP, Dnhe2* [68.4 ± 1.2, n = 4]). (*$p < 0.05$, **$p < 0.01$, ***$p < 0.001$).

imaginal discs termed the second mitotic wave (SMW) is present posterior to the morphogenetic furrow, a physical indentation in the epithelium that serves as a spatial and temporal benchmark of eye development. Posterior to the SMW, cells divide stochastically. With *GMR > Dnhe2* expression, we found a significant increase in the number of proliferating phospho-histone H3 (phosH3) positive cells within this posterior region (*Figure 2C,D*). We also expressed *Dnhe2* in the posterior compartment of the developing wing imaginal disc using an *engrailed* driver (*enGAL4*). In *en > GFP, Dnhe2* expressing wing discs, there was overgrowth and dysplasia only in the expressing compartment, while non-expressing tissue resembled wild type (*Figure 2E*). In wild type wing discs, the number of phosH3 positive cells was relatively similar in the posterior and anterior compartments (*Figure 2F,G*). With *en > Dnhe2* expression, however, this distribution was markedly different with significantly more proliferating cells in the posterior compared with anterior compartment (*Figure 2F,G*). Together, these data suggest that *Dnhe2* acts in a cell-autonomous manner to promote proliferation and dysplasia.

NHE1 activity promotes directed cell migration (*Denker and Barber, 2002*; *Patel and Barber, 2005*; *Stock and Schwab, 2009*) and invasion (*Stock and Schwab, 2009*; *Busco et al., 2010*) in cultured cells; however effects on these behaviors in vivo remain unresolved. We tested for synergy between *Dnhe2* and oncogenic *RasV12* using the *ptc-GAL4* driver to express transgenes in a stripe of cells along the anterior/posterior compartment boundary in wL3 wing discs; this assay induces mosaic transgene expression to distinguish autonomous and non-autonomous effects, and can reveal effects on tissue organization, proliferation and invasive/migratory cell behaviors (*Vidal et al., 2006*; *Petzoldt et al., 2013*) (*Figure 3*). Expression of *RasV12* induced a wider *ptc > GFP* stripe compared to control (~1.6-fold increase), while expression of *Dnhe2* alone did not (*Figure 3A–C*). However, co-expression of *RasV12* and *Dnhe2* markedly increased the *ptc > GFP* stripe width greater than twofold (*Figure 3C*). We also observed a significant increase in the overall size of the wing disc with *ptc > RasV12, Dnhe2* compared with either transgene alone or wild type (*Figure 3A,D*). To determine invasive phenotypes, we identified single GFP-positive cells in stacks of confocal images. In control and *Dnhe2*-expressing discs we found only a single instance of a few (*Rofstad et al., 2006*; *Stock and Schwab, 2009*; *Zhang et al., 2010*) single GFP-positive cells, and no examples in *RasV12*-expressing discs (*Figure 3E*). However, in *ptc > RasV12, Dnhe2* discs, we found examples of invasive cells in all discs examined. Further, we found several different types of invasive behaviors, including basal expansion of the *ptc* stripe, single invasive cells and streams of cells migrating out of the stripe into neighboring tissue in the posterior compartment (*Figure 3E*). These data show cell autonomous effects of *Dnhe2* expression on growth and migration behaviors, as well as non-autonomous effects on tissue growth.

To address our second question on whether H$^+$ efflux is necessary for oncogene-induced dysplasia, we generated precise deletions of the *Dnhe2* using FLP-mediated recombination of FRT-bearing transposon insertions that flank the *Dnhe2* coding region (*Figure 4A,B*). This strategy avoided excision of a neighboring gene CG9257 that caused larval lethality in the existing *Dnhe2[1]* allele (*Simons et al., 2009*). Deletions were screened by PCR by using primer pairs to test for the presence of coding sequences as well as for the recombinant P-element generated by recombination (*Figure 4B*). Five independent excision lines were obtained, and all showed identical phenotypes. Homozygous *Dnhe2[null]* flies had ~20% survival to eclosion (*Figure 4C*), and surviving adult flies appeared morphologically normal. The lethality of *Dnhe2[null]* flies was rescued by expression of *Dnhe2* using the muscle-specific *mef2-GAL4* driver, but not using the neuronal *elav-GAL4* driver (*Figure 4C*).

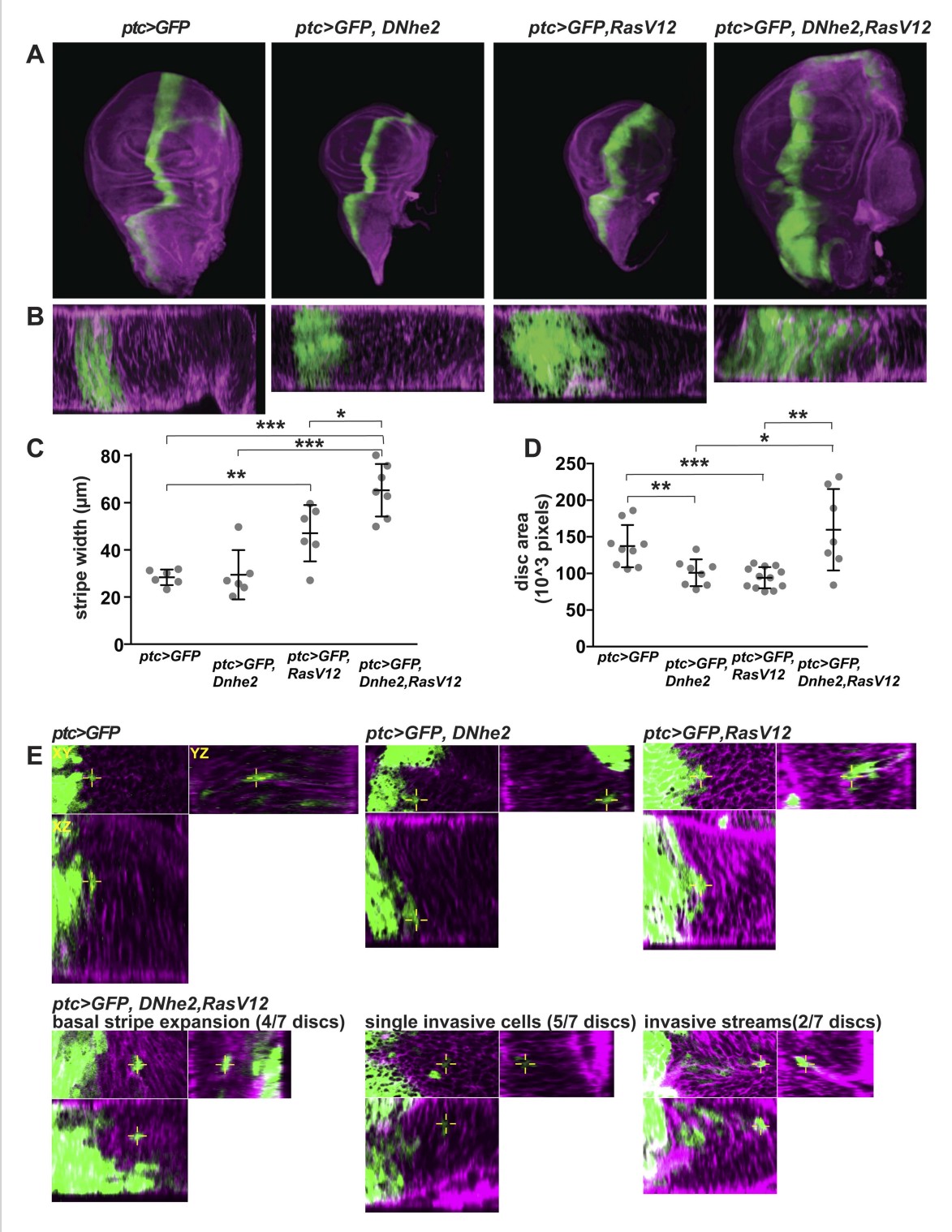

**Figure 3**. *Dnhe2* co-expression enhances RasV12-mediated growth and invasion. (**A**) Confocal images of third instar wing imaginal discs expressing *ptc > GFP* plus indicated transgenes (green) and rhodamine-phalloidin (purple) fluorescence in XY maximum projection images and (**B**) XZ single slices below (posterior to the right). (**C**) Quantification of *ptc > GFP* stripe width (in μm; mean ± SEM: *ptc > GFP* [26.4 ± 1.3, n = 6], *ptc > GFP, Dnhe2* [27.4 ± 4.0, n = 6], *ptc > GFP, RasV12* [43.8 ± 4.5, n = 6], *ptc > GFP, RasV12, Dnhe2* [60.7 ± 3.9, n = 7]). (**D**) Quantification of total area of wing imaginal discs (in 10³ pixels; mean ± SEM, *ptc > GFP* [137.3 ± 9.6, n = 9], *ptc > GFP, Dnhe2* [107.8 ± 8.9, n = 9], *ptc > GFP, RasV12* [93.5 ± 4.1, n = 12], *ptc > GFP, RasV12, Dnhe2* [196.5 ± 41.4, n = 8]). (**E**) Invasive cells were identified as GFP-positive cells that were isolated from other GFP-positive cells in three-dimensional tissue architecture. For each example, shown are XY (upper left), YZ (upper right) and XZ (lower left) single confocal slices. Invasive phenotypes were rarely seen
*Figure 3. continued on next page*

Figure 3. Continued

in *ptc > GFP*, *ptc > GFP, Dnhe2* discs, or *ptc > GFP, RasV12* wing discs. In *ptc > GFP, RasV12, Dnhe2* discs, examples of basal stripe expansion (4/7), multiple single, invasive cells (5/7 discs) and streams of invading cells (2/7) were seen. (*p < 0.05, **p < 0.01, ***p < 0.001).

To test the effects of deleting *Dnhe2* with expression of an activated oncogene we initially used *GMR-GAL4* to drive activated *RasV12*; however *GMR > RasV12* caused pupal lethality, which precluded phenotypic analysis in adult flies. We therefore induced expression of constitutively active, oncogenic *Raf* (**Brand and Perrimon, 1994**) (*GMR > Raf*). Retinal architecture in homozygous *Dnhe2^null^* flies was organized and regular, resembling wild type (**Figure 5A,B**). The external surface of *GMR > Raf* eyes was disorganized with indistinct facets and a rough exterior retinal surface (**Figure 5A,B**). Some ommatidia showed thin or absent lenses. In section, *GMR > Raf* had ommatidial defects with mis-orientation and occasional missing rhabdomeres (**Figure 5C**, tracing in **Figure 5D**). Additionally, *GMR > Raf* rhabdomeres were morphologically distinct from wild type, appearing more rectangular, especially in R1, R3, R5 and R6. Externally, *Dnhe2^null^; GMR > Raf* appeared identical to *GMR > Raf* (**Figure 5A,B**), except that the eye color was noticeably lighter and yellowed. Tangential retinal sections revealed a complete loss of tissue architecture and cell structure, with histological features of coagulative necrosis, including loss of rhabdomere morphology, absence of distinct cell membranes and diffusion of pigment granules (**Figure 5C**). Live, dissected whole-mount adult eyes revealed a significant reduction of the cohesive, red-pigmented retinal epithelial layer, such that the remaining *Dnhe2^null^; GMR > Raf* tissue was thin and transparent in appearance (data not shown). These observations support a synthetic lethal interaction between loss of *Dnhe2* and expression of oncogenic Raf.

Synthetic lethality of *Dnhe2^null^; GMR > Raf* was progressive, as determined by examining earlier developmental stages. In wL3 eye discs, *Dnhe2^null^* tissue morphology was indistinguishable from wild type (**Figure 6A**); however there were fewer proliferating cells than in wild type as indicated by phosH3 staining (**Figure 6A,B**). Expression of *GMR > Raf* led to disrupted tissue morphology, with irregular ommatidial spacing and folding of the eye disc, presumably due to increased proliferation (**Figure 6A,B**). Compared with *GMR > Raf* the morphology of *Dnhe2^null^; GMR > Raf* appeared less disrupted (**Figure 6A**) and hyperproliferation was suppressed (**Figure 6B**). Pupal eyes were assayed 42H apf for expression of the neuronal antigen Elav, and we found that cells in all genotypes retain their fates as differentiated neurons (**Figure 6C**). Beta-catenin immunolabeling showed distinct differences in the morphology of cell contacts (**Figure 6D**, line drawing in **Figure 6E**). *Dnhe2^null^* pupal eyes appeared wild type: individual ommatidia had 4 cone cells and 2 primary pigment cells visible on the apical surface, and a single row of secondary pigment cells separating neighboring ommatidia. However, cell morphologies were slightly abnormal (**Figure 6D,E**). Secondary pigment cells were thicker than in wild type, and cell shapes and cell–cell contacts were irregular. With expression of *GMR > Raf*, cell morphologies were profoundly aberrant, with extra and mis-shapen cone cells and ommatidia mis-aligned along the D/V axis, as seen in adults. Additionally, immature cone cell contacts were maintained and secondary pigment cells were absent between some pairs of neighboring ommatidia. These defects were enhanced in *Dnhe2^null^; GMR > Raf* pupal eyes, where all cell types appeared smaller and more cells had abnormal morphologies. Additionally, there were more ommatidia with extra cone cells. Immunolabeling pupal eyes for cleaved caspase 3 (CC3), a marker of apoptotic cells, showed no staining in wild type, *Dnhe2^null^* or *GMR > Raf* eyes. However, in *Dnhe2^null^; GMR > Raf* eyes there were a few CC3–positive cells in all retina examined, even though the normal phase of cell death concludes ~12 hr before this time point (data not shown). CC3 staining was not abundant enough to indicate widespread cell death at 42H apf. Our data suggest that *Dnhe2* deletion causes progressive defects in Raf-expressing cells, which culminates in synthetic lethality prior to eclosion.

Changes in pHi correlated with the progressive morphological defects described above. We measured pHi using the mCherry-pHluorin biosensor in live wL3 eye imaginal discs, approximately 24 hr after the onset of oncogenic *Raf* expression (**Figure 7A**). Compared with wild type, the pHi in *Dnhe2^null^* eye discs was significantly lower and *GMR > Raf* was significantly higher (**Figure 7B**). In contrast, pHi in *Dnhe2^null^; GMR > Raf* discs was not different than wild type (p > 0.05). This suggests that the increased pHi with expression of oncogenic Raf requires *Dnhe2* activity. We also measured

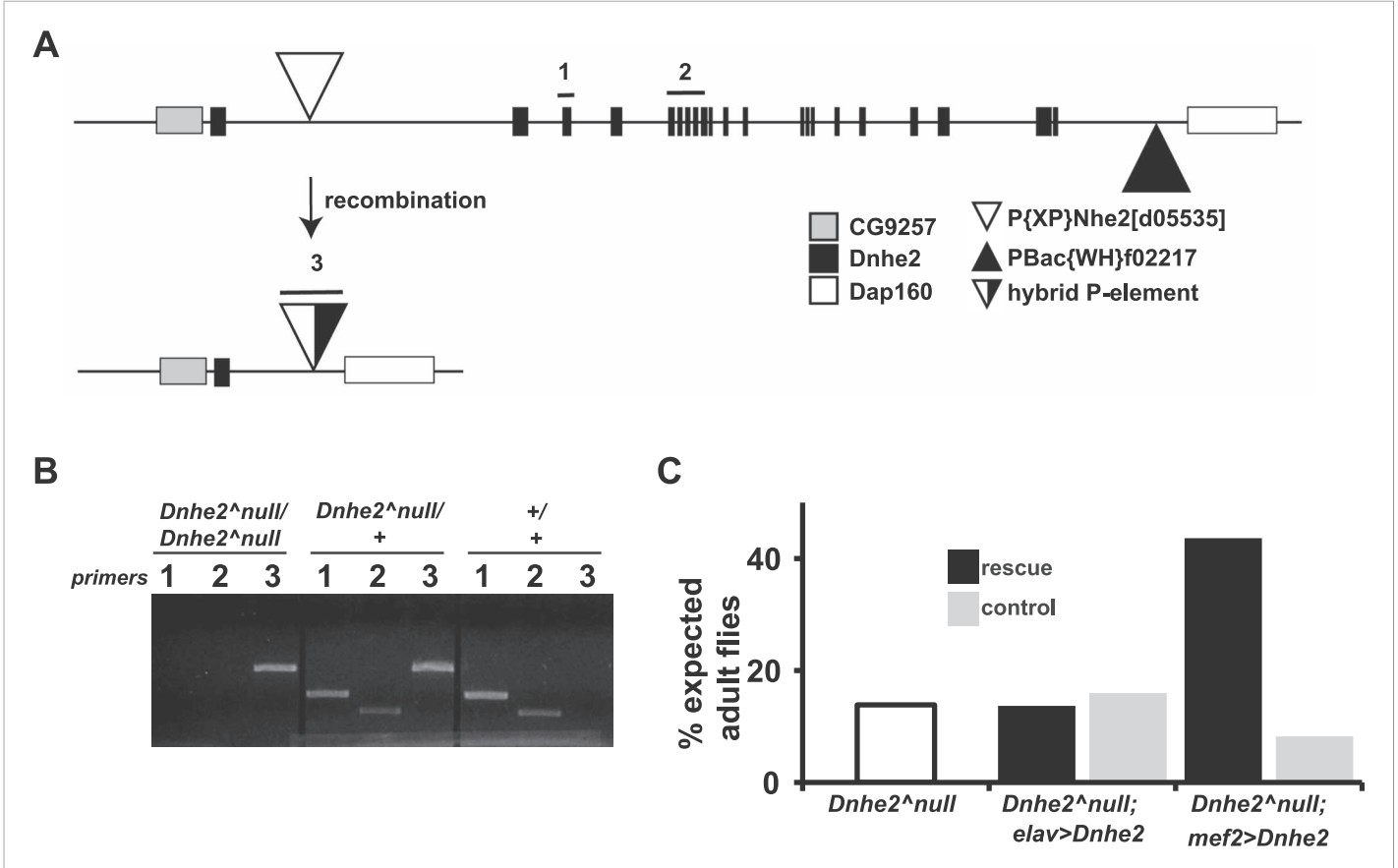

**Figure 4**. Deletion of Dnhe2 is semi-lethal with a genetic requirement in muscle. (**A**) Schematic diagram of *Dnhe2* genomic locus, showing *Dnhe2* exons (black rectangles), neighboring genes CG9257 and Dap160 (grey and white rectangles, respectively) and P-elements P{XP}Nhe2[d05535] (white triangle) and PBac{WH}f02217 (black triangle). Following FLPase-mediated recombination, all coding exons of *Dnhe2* were excised, leaving a hybrid P-element (black and white triangle). PCR primer pairs 1 and 2 are within *Dnhe2* coding sequence, while primer pair 3 covers the recombinant P-element. (**B**) Sample PCR reactions from genomic DNA preps isolated from homozygous *Dnhe2^null*, heterozygous and wild type adult Drosophila. Primer pairs 1 and 2 yield product only in wild type or heterozygotes. Primer pair 3 is generated in *Dnhe2^null* homozygotes or heterozygotes. (**C**) *Dnhe2^null* flies show 17.7% survival to adulthood. Rescue experiments that restore *Dnhe2* expression in neurons (*Dnhe2^null*; *elavGAL4/UASDnhe2*) do not rescue (13.7%), but expression in muscles (*Dnhe2^null*; *mef2GAL4/UASDnhe2*) rescues to 43.6% survival. Controls (GAL4 driver alone) do not show any rescue effects with 16.0% survival for *Dnhe2^null*; *elavGAL4/+* and 8.2% survival for *Dnhe2^null*; *mef2GAL4/+*.

pHi in live 42H apf pupal eyes, and found that pHi in *GMR > Raf* was still significantly higher than wild type pupal eyes (*Figure 7C,D*). In contrast to larval eye discs, however, the pHi of *Dnhe2^null*; *GMR > Raf* was significantly lower than wild type and lower than *GMR > Raf* alone (*Figure 7C,D*). These data suggest that pHi progressively decreases through pupal morphogenesis in *Dnhe2^null*; *GMR > Raf* retinae, and this could contribute to the synthetic lethality observed in adult eyes.

We also found synthetic lethality in clonal cells with loss of H⁺ efflux and oncogene expression. In clonal human mammary MCF10A cells we found that loss of NHE1 activity lowers pHi and increases cell death selectively in cells expressing oncogenic RasV12. We first used an estrogen receptor-induced H-RasV12 (ER-RasV12) and suppressed NHE1 expression with shRNA. Tamoxifen-induced Ras expression markedly increased rate of pHi recovery from an acid load (*Figure 8A*, *Figure 8—figure supplement 1*). Tamoxifen increased pHi in ER-Ras cells and with NT shRNA but not in vector control cells (*Figure 8B*). However, with NHE1 shRNA pHi was significantly lower (*Figure 8B*), indicating NHE1 is necessary for increased pHi with oncogenic Ras. Tamoxifen treatment increased NHE1 expression (*Figure 8C*), while NHE1 shRNA treatment significantly reduced NHE1 protein expression (*Figure 8C*; two different NHE1 shRNA constructs showed similar results). A similar NHE1-dependent increase in pHi was shown using MCF10A cells stably expressing H-RasV12 (sRasV12) and 5(N-ethyl-N-isopropyl)

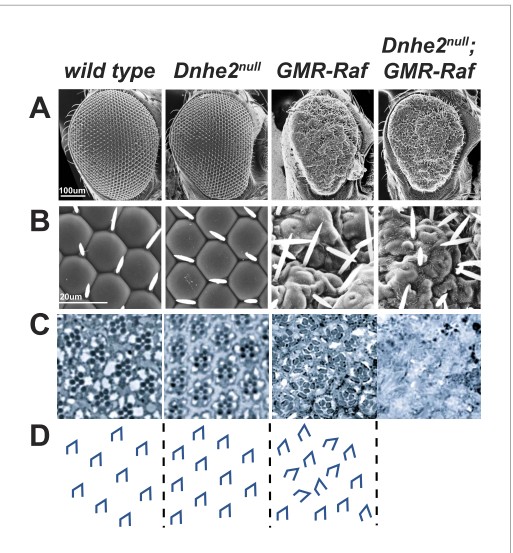

**Figure 5**. Synthetic lethality with oncogenic Raf and loss of *Dnhe2*. (**A** and **B**) Scanning electron micrographs of the exterior surface of adult Drosophila eyes. Expression of *GMR > Raf* causes a rough eye phenotype in the absence and presence of *Dnhe2*. (**C**) Histological sections reveal a trapezoidal arrangement of rhabdomeres in wild type and *Dnhe2^null* adult eyes. *GMR > Raf* eyes show ommatidial rotation defects and abnormal photoreceptor recruitment. Histological sections through *Dnhe2^null*; *GMR > Raf* eyes show a complete absence of tissue organization consistent with coagulative necrosis. (**D**) Traced diagrams show ommatidial orientation.

amiloride (EIPA) to pharmacologically inhibit NHE1 activity (*Figure 8E,F*). As a control, we show that EIPA treatment does not effect Ras expression (*Figure 8G*). Although pHi was higher in MCF10A cells with RasV12 and inhibited NHE1 activity than in *Drosophila Dnhe2^null*; *GMR > Raf* pupal eyes, measuring pHi of cultured cells requires washes that could remove less adherent, dying cells with a lower pHi. Quantifying dying cells by flow cytometry using fluorescently labeled AnnexinV showed NHE1 shRNA or EIPA significantly increased cell death with RasV12 ($p < 0.05$) (*Figure 8D,H*) but not in control cells. These results combined with our in vivo findings in *Drosophila* indicate that limiting $H^+$ efflux causes synthetic lethality with oncogene expression.

Decreasing $H^+$ efflux also increased lethality of invasive tumor cells having a complex mutational landscape compared with only H-RasV12 expression as described above. We used SW620 metastatic colorectal cells, which have an activating mutation in K-Ras (G12V), mutations in the tumor suppressors SMAD4 (frameshift), p53 (R273H, P309S), and adenomatous polyposis coli (APC) (Q1338*), as well as MDA-MB-231 invasive breast metastatic cells, which have a different activating mutation in K-ras (G13D), a mutant B-Raf (G464V) and mutations in the tumor suppressors cyclin-dependent kinase inhibitor 2A (deletion), and p53 (R280K). In both cell lines, EIPA treatment for 72 hr significantly decreased pHi (*Figure 9A*) and increased cell death compared with controls (*Figure 9B*). These data indicate that loss of $H^+$ efflux limits survival of cells containing multiple oncogenic lesions and suggest a broad applicability of limiting $H^+$ efflux to increase cell death across cancer types.

## Discussion

With the advent of low-cost whole genome sequencing, personalized chemotherapeutic treatments currently dominate discussion of new cancer therapies. However, the balkanization of cancers by causative mutation or tissue of origin ignores the shared disease physiology in most cancers as highlighted by the 'hallmarks of cancer' (*Hanahan and Weinberg, 2011*). Dysregulated pHi is an emerging hallmark of cancer that is associated with traditional hallmarks, including resisting cell death, sustaining proliferative signaling, activating invasion and metastasis, deregulating cellular energetics and tumor-promoting inflammation. Therefore, targeting pHi-regulatory proteins in cancer cells could address distinct pathological characteristics in parallel.

Our data provide an in vivo functional corollary of the established increased pHi and upregulation of $H^+$ efflux mechanisms in cancer cells. Building on previous reports that NHE1 activity and expression is increased in human cancers (*McLean et al., 2000*; *Miraglia et al., 2005*; *Chiang et al., 2008*; *Yang et al., 2011*), we show that engineered over-expression of an NHE1 ortholog in vivo increased pHi by ~0.3 pH units, which is comparable to the increase seen with oncogene transformation. Moreover, over-expression of *Dnhe2* was sufficient to induce dysplasia and hyper-proliferation phenotypes, and it enhanced oncogene-mediated invasion. Further, co-expressing *RasV12* and *Dnhe2* had a strong, synergistic effect on growth and invasion. A similar synergistic effect between expression of *RasV12* and the C subunit of the V-ATPase, *Vha44* was recently reported (*Petzoldt et al., 2013*), which supports the ability of increased $H^+$ efflux to enhance RasV12-induced phenotypes. However, *Vha44* enhances invasion with RasV12 but not activated Src or Abl, suggesting an oncogene-specific synergy. Important

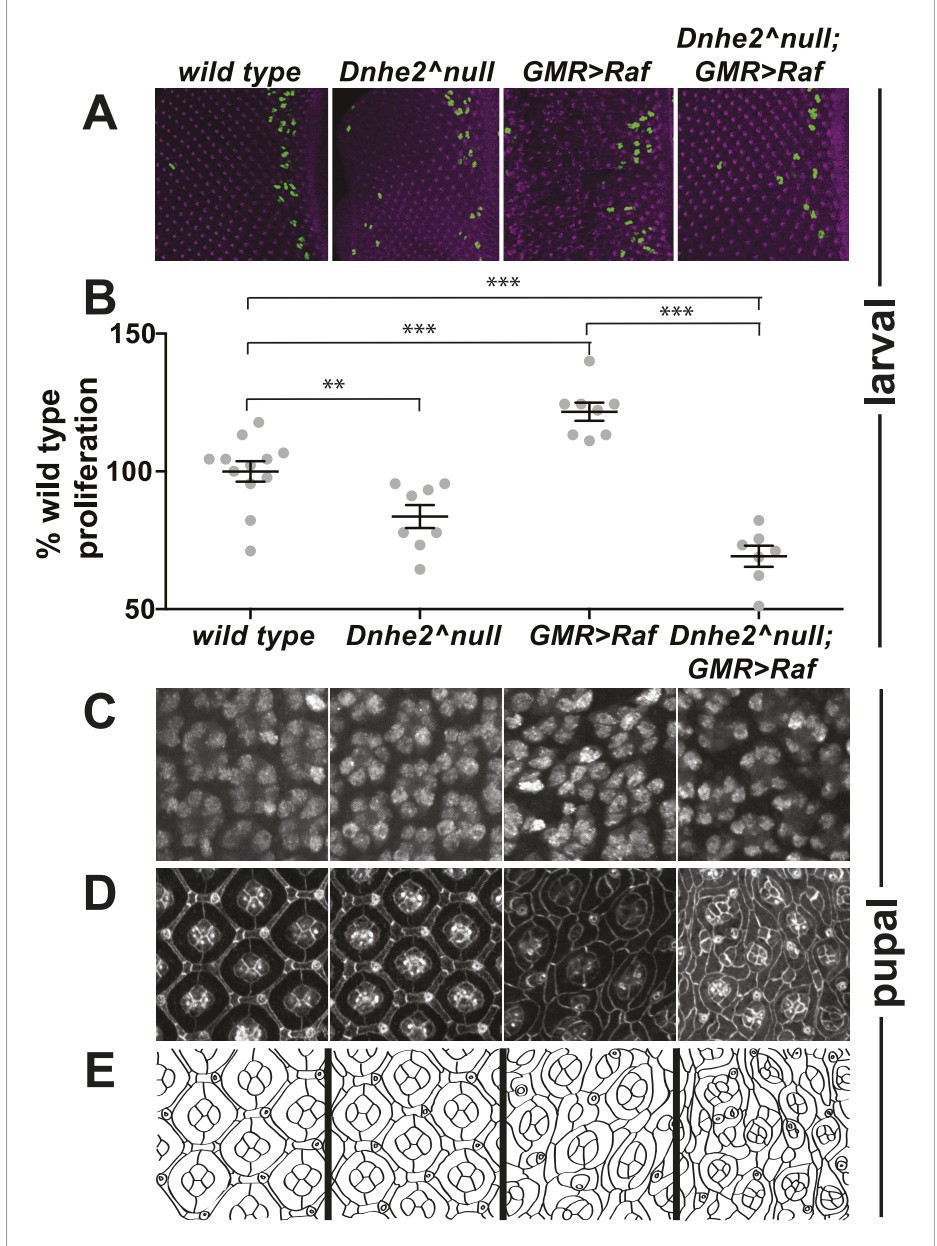

**Figure 6**. Progressive synthetic defects with oncogenic Raf and loss of *Dnhe2*. Phenotypic analyses are shown for the indicated genotypes at third larval instar (wL3) and mid-pupal developmental stages. (**A**) wL3 eye discs labeled for phospho-Histone3 (green) to indicate dividing cells, and E-cadherin to show adherens junctions (purple). (**B**) The number of phos-H3 cells posterior to the morphogenetic furrow were counted, and shown as % of wild type (mean ± SEM): *w1118* (n = 12), *Dnhe2^null* (83.6 ± 4.2, n = 8), *GMR > Raf* (121.7 ± 3.3, n = 8), *Dnhe2^null*; *GMR > Raf* (68.2 ± 3.8, n = 7). (**C**) In pupal eyes, Elav labels photoreceptor neurons in all genotypes. (**D**) In 42H apf pupal eyes, adherens junctions are labeled with βeta-catenin, with schematic drawing shown in (**E**). Individual ommatidia have four central cone cells that are enwrapped by two primary pigment cells. These ommatidial cores are insulated from each other by a single row of interommatidial cells (IOCs). This organization is disrupted with expression of *GMR > Raf*, and further enhanced with deletion of *Dnhe2*. (*p < 0.05, **p < 0.01, ***p < 0.001).

to determine in future studies are the relative effects of increased pHi on growth and invasion, and whether decreased pHe contributes to the non-autonomous effects we observed in imaginal discs.

Our data also indicate that reducing H+ efflux in cancer cells suppresses multiple pathological characteristics in parallel. Previous studies identified distinct responses to reduced H+ efflux, such as

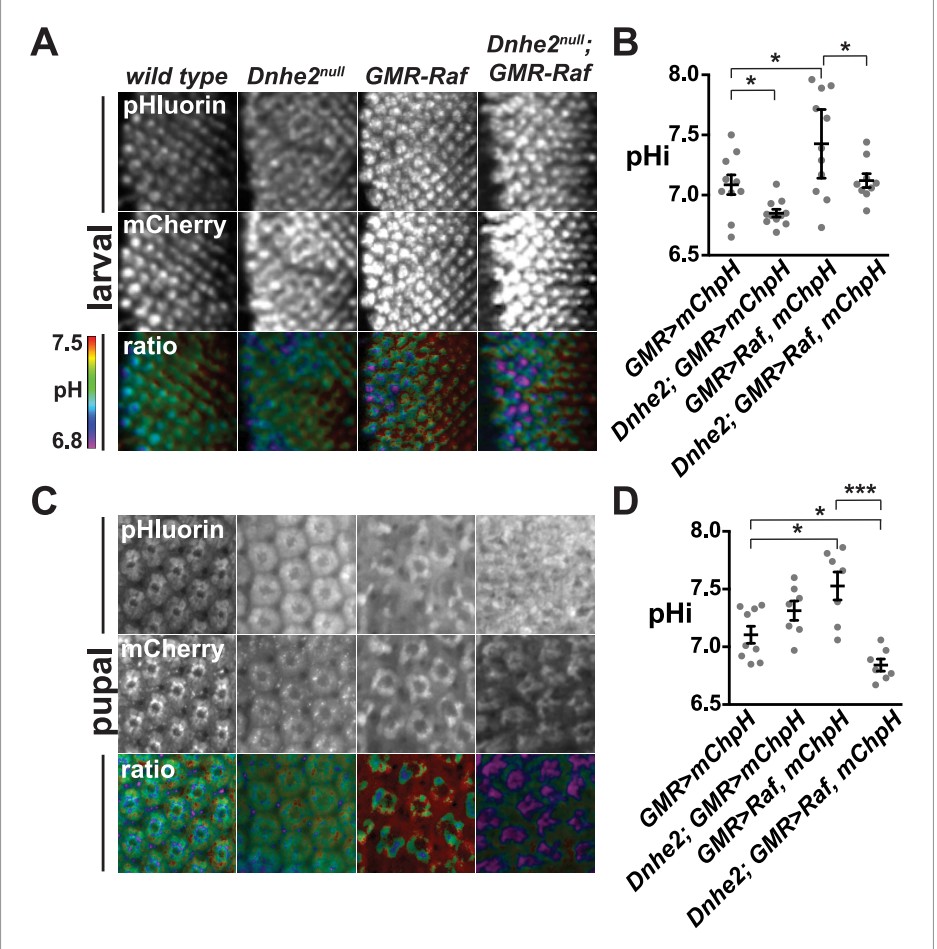

**Figure 7**. *Dnhe2* deletion progressively decreases pHi in Raf-expressing but not wild type cells. pHi was determined using *GMR > mCh-pH*. (**A**) Single confocal slices from live, third larval instar eye imaginal discs showing fluorescence of pHluorin (top row), mCherry (middle row), and ratiometric images (bottom row). Ratiometric images were generated by dividing the fluorescence intensity of pHluorin by mCherry. (**B**) pHi was calculated from ratio values using a standard curve and graphed (mean ± SEM). Estimated pHi values in third larval instar retinal tissue are: wild type (7.1 ± 0.08, n = 10); *Dnhe2null* (6.9 ± 0.03, n = 11); *GMR > Raf* (7.4 ± 0.13, n = 11); *Dnhe2null*; *GMR > Raf* (7.1 ± 0.06, n = 9). (**C**) Single confocal slices from live, pupal eyes, as described above. (**D**) Estimated pHi values in pupal retinal tissue are: wild type (7.1 ± 0.07, n = 9), *Dnhe2null* (7.3 ± 0.08, n = 7); *GMR > Raf* (7.5 ± 0.12, n = 7); *Dnhe2null*; *GMR > Raf* (6.8 ± 0.05, n = 7). (*p < 0.05, ***p < 0.001).

tumor growth in xenograft models (*Lagarde et al., 1988*; *Sonveaux et al., 2008*; *Colen et al., 2011*; *Yang et al., 2011*) or viability of isolated cancer cells (*Rich et al., 2000*; *Reshkin et al., 2003*). We now show progressive effects of limiting H⁺ efflux with oncogene expression, including initially suppressing hyper-proliferation and dysplasia, followed by decreasing pHi, and culminating in a synthetic lethal interaction with histological features of coagulative necrosis. Coagulative necrosis is characteristically found within the central region of solid tumors (*Searle et al., 1975*; *Lagarde et al., 1988*), and is thought to be induced by extreme physiological cellular stresses. Removing *Dnhe2* in an otherwise wild type genetic background decreased pHi but had no effect on larval tissue morphology. *Dnhe2null* retinal pHi in pupae was slightly higher than in larvae but not significantly different compared with wild type. These data suggest that in the absence of *Dnhe2*, pHi homeostasis during metamorphosis can be maintained by alternative ion transport mechanisms, changes in cellular buffering capacity and/or metabolic changes. While our *Drosophila* studies evaluate synthetic lethality in the ~20% *Dnhe2* homozygous mutant 'escaper' flies that survive to eclosion, our studies in several transformed cell lines also show a synthetic lethality interaction, suggesting a conserved mechanism. The synthetic

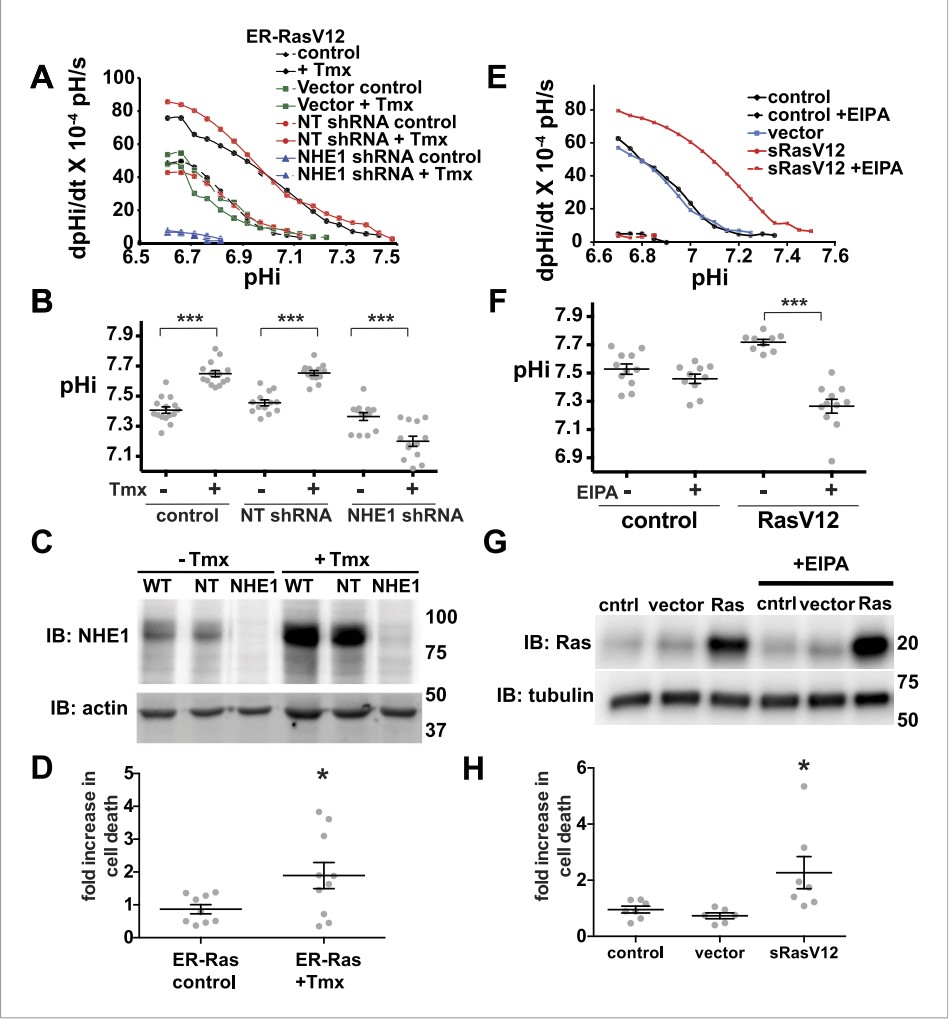

**Figure 8**. Decreased NHE1 expression or activity with RasV12 expression prevents increased pHi in human MCF10A cells and shows synthetic lethality. (**A**) In MCF10A cells expressing ER-RasV12 the rate of pHi recovery from an acid load, an index of H⁺ efflux, is substantially greater in tamoxifen (Tmx)-treated controls (solid black line) and with non-targeting (NT) shRNA (solid blue line) than in the absence of Tmx (dotted black and blue lines, respectively) or in vector controls (solid and dotted green lines). In NHE1 shRNA cells pHi recoveries are nearly abolished in the absence (dotted red line) or presence (red solid line) of Tmx. (**B**) Tamoxifen increases steady-state pHi in control (mean ± SEM for all condition: control 7.41 ± 0.02, +Tmx 7.65 ± 0.02) and NT shRNA control (7.45 ± 0.02, +Tmx 7.65 ± 0.02) cells but not in NHE1 shRNA cells (7.36 ± 0.03, +Tmx 7.20 ± 0.03). (**C**) Expression of NHE1 is increased upon tamoxifen (Tmx) treatment of ER-RasV12 cells. Treatment of cells with NHE1 shRNA decreases NHE1 expression. (**D**) Cell death, indicated by Annexin V-positive cells is significantly greater with NHE1 shRNA in ER-RasV12 expressing cells (1.9× increase over NT shRNA), but not in control cells (0.87×). (**E**) The rate of pHi recovery from an acid load is substantially greater in MCF10A cells stably expressing RasV12 (sRasV12, red solid line) compared with control (WT, solid black line) or vector expression (solid blue line). The NHE1 inhibitor EIPA abolishes pHi recoveries in control (dotted black line) and sRasV12 cells (dotted red line). (**F**) Steady-state pHi is significantly higher in sRasV12 cells compared with control cells. With EIPA treatment, steady-state pHi is significantly lower in sRasV12 (control 7.72 ± 0.02, +EIPA 7.27 ± 0.02) but not in control cells (control 7.53 ± 0.03, +EIPA 7.46 ± 0.03). (**G**) sRasV12 expressing cells show increased expression of Ras compared to either control or vector cells. Ras expression is unaffected by inhibition of NHE1. (**H**) Treatment with EIPA induces cell death only in sRasV12 expressing cells (2.3× increase over control), but not in vector (0.73×) or control (0.95×) cells. Sample were pooled from three independent cell preparations (for NHE1 shRNA, two different shRNA constructs were tested in two cell preparations and pooled). (*p < 0.05, **p < 0.01).

The following figure supplement is available for figure 8:

**Figure supplement 1**. RasV12 expression increases NHE1 activity and pHi.

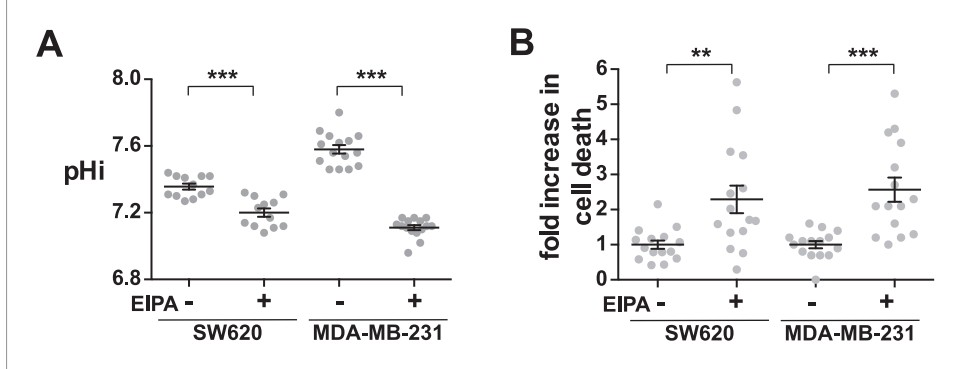

**Figure 9**. Decreased NHE1 activity in tumor-derived cell lines induces cell death. (**A**) pHi is decreased with EIPA treatment in both SW620 metastatic colorectal tumor cells (control 7.36 ± 0.02, +EIPA 7.2 ± 0.03), and MDA-MB-231 metastatic breast cells (control 7.58 ± 0.03, +EIPA 7.11 ± 0.01). (**B**) Cell death as determined by a trypan blue exclusion assay is also increased in both cell lines with EIPA treatment over control: SW620 2.3×; MDA-MB-231 2.6×. Data are from four or five independent cell preps. (*p < 0.05, **p < 0.01, ***p < 0.001).

lethal interaction we observed suggests $H^+$ efflux is necessary for dissipating oncogene-generated acids, most likely from increased glycolytic metabolism. Our studies further show that these lethal interactions occur in two genetically complex, physiologically adapted invasive tumor cell lines.

Our data build on the long-standing observation of increased pHi in cancer cells and suggest several new directions for future investigation. First, our findings indicate that the biosensor pHluorin can be used to resolve pHi dynamics in vivo during tumor formation and metastasis and whether there are spatially distinct pHi dynamics within a tumor that might inform us about tumor properties and plasticity. Second, the rough eye phenotype induced with *GMR > Dnhe2* provides a new model for genetic screens to identify pH-sensitive pathways and previously unrecognized pH sensors regulating cell growth and tissue architecture. Dysplasia with increased pHi in the absence of oncogenes is likely in part dependent on cancer-promoting pH sensors (*Schönichen et al., 2013*), including focal adhesion kinase (FAK) and the actin regulatory protein cofilin that support tumor growth and metastasis (*Wang et al., 2007*; *Cance et al., 2013*) and have increased activity at higher pHi (*Frantz et al., 2008*; *Choi et al., 2013*). Third, the synthetic lethality we show suggests that therapeutic strategies targeting $H^+$ efflux transporters or pH sensors with selective roles in oncogene-induced responses could limit metastatic progression with minimal off-target effects.

## Materials and methods

### Fly stocks

Standard *Drosophila* genetic techniques were used. Flies were cultured at 25°C unless otherwise noted. Stocks were obtained from the Bloomington *Drosophila* Stock Center: *UAS-RasV12* (4847), *w1118* (5905), *mef2-GAL4* (27,390), *eyGAL4* (8219, 8220), *UAS-Raf* (2033); or the Harvard stock center: P(XP) Nhe2d05535 and PBac(WH)f02217. Other stocks used were *GMR-GAL4* (gift from R Cagan) and *enGAL4* and *ptc-GAL4*, *UAS-GFP* (gifts from D Casso).

### *Dnhe2, Dnhe2E358I, and mCherry-pHluorin* transgenic lines

*Dnhe2* was originally cloned from a *Drosophila* S2 cell-derived cDNA library (GenBank AF235935). We isolated a 3.7 kb *Dnhe2* cDNA from S2 cells that is essentially identical to the C isoform predicted in Flybase, with the following polymorphisms: A157T, T283I, insL568, missing exon 17 (72 amino acids), 12 additional amino acids at the C terminus (SKGEFQHTGGRY). *Dnhe2* cDNA was cloned into pUAST, and transgenic flies were generated (BestGene, Inc., Chino Hills, CA) and reported here as *UAS-Dnhe2*.

To generate *UAS-Dnhe2E358I*, the cDNA was amplified by PCR, adding a 5' KpnI site and a 3' HA-tag followed by an ApaI restriction site, and subcloned into pCR-TOPOXL. The E358I mutation was generated using Invitrogen QuikChange XL Site Directed Mutagenesis Kit and the following primers:

Nhe2 E358I F: gtcgtctttgggatatccttgctgaacgatgccgtcacggttg.

Nhe2 E358I R: caaccgtgacggcatcgttcagcaaggatatcccaaagacgac.

The mutagenesis also introduced an EcoRV restriction site to facilitate colony screening.

pEGFP-mCherry-pHluorin was obtained from S Grinstein (*Koivusalo et al., 2010*). pEGFP-mCherry-pHluorin was digested with EcoRI and NotI, and a ~1.5 kb mCherry-pHluorin containing fragment was gel purified and ligated into pUAST. Transgenic flies were generated (BestGene, Inc.), and are herein referred to as *UAS-mCherry-pHluorin*.

To generate *Dnhe2null* alleles, FLP-FRT recombination was used to generate w⁻ deficiencies from Exelixis transposon insertions P(XP)Nhe2d05535 and PBac(WH)f02217. Excisions were further screened by PCR for the presence of the resulting hybrid residual element (XP:WH) by PCR using element specific primers as described (51A and 5A1) (*Parks et al., 2004*). Lethality was calculated as percentage of expected homozygous null flies to eclose as adults (number of flies scored: *Dnhe2null* [1050]; *Dnhe2null*; *mef2GAL4/UASDnhe2* [1231]).

## Scanning electron microscopy

Adult heads were removed, bisected sagittally and fixed overnight in 2% glutaraldehyde + 4% paraformaldehyde in 0.1 M sodium phosphate buffer, pH 7.4. Samples were processed in a critical point dryer, mounted on carbon tape (Ted Pella, Redding, CA 16,084–6) or with colloidal silver (Ted Pella, 16,034) on an SEM stub (Ted Pella, 16,221) and splutter coated with gold-palladium (Ted Pella, 22–2). Images were collected on a JEOL Neoscope Scanning Electron Microscope, and cropped in Photoshop.

## Tissue histology

Pupal and larval retinal and wing tissue were dissected in PBS, fixed in fresh 4% paraformaldehyde, washed in PBS + 0.1% Triton X-100, immunostained and mounted in N-propyl gallate or Prolong Gold (Life Technologies, Grand Island, NY, P36930). Antibodies used were: mouse anti-beta-catenin (cat# N2 7A1 ARMADILLO; 1:10), rat anti-ELAV (cat# Rat-Elav-7E8A10; 1:20), rat anti-Ecadherin (cat# DCAD2; 1: 10), rat anti-Elav (cat# Rat-Elav-7E8A10; 1:20), Developmental Studies Hybridoma Bank, Iowa City, IO; rabbit anti-cleaved-caspase 3 (cat# 9661, 1:100), mouse anti-phospho-histone 3 (cat# 9706, 1:100), Cell Signaling Technologies, Danvers, MA. In *Figure 1*, cells were scored as posterior to the SMW if they lay posterior to the 10th ommatidial row behind the morphogenetic furrow.

Continuous *ptcGAL4*-mediated expression of either *RasV12* or *Dnhe2* induced early larval lethality, therefore flies were raised at 18°C to limit transgene expression until the L2/L3 transition, when they were shifted to 25°C to permit transgene expression for 48 hr prior to dissection. *ptc-GFP* stripe widths were measured at the dorsal-ventral boundary. To identify invasive cells, confocal Z-stack images were analyzed in three dimensions using NIS Elements software (Nikon). When potential invasive cells were identified, the location was marked with a crosshair, and XZ and YZ projections were examined to determine whether these GFP-positive invasive cells were isolated in three-dimensional space.

Adult tissue dissection, fixation and resin embedding were performed as described (*Wolff, 2000*). Tissue was cut into 0.5 μm sections on a microtome and contrast stained with toluidine blue. Phase contrast images were collected on a Zeiss Axiophot microscope using a 63× oil objective and a Hamamatsu ORCA-ER digital camera. Images were cropped, pseudo-colored and brightness/contrast adjusted in Photoshop.

## In situ pHi measurements

Transgenic *Drosophila* lines expressing the pH sensor soluble mCherry-pHluorin (*Koivusalo et al., 2010*) (*UAS-mCherry-pHluorin*) were crossed to *GMR-GAL4* to express the pH biosensor specifically in the developing eye. pH measurement were determined as described (*Grillo-Hill et al., 2014*). Briefly, retinae were dissected from wandering third instar larvae or from 42H apf pupae in HCO₃ buffer (25 mM NaHCO₃, 115 mM NaCl, 5 mM KCl, 10 mM glucose, 1 mM MgSO₄, 1 mM KHPO₄, 2 mM CaCl₂) and mounted in MatTek glass-bottom dishes. Fluorescent images were collected on an inverted spinning disk confocal microscope (*Stehbens et al., 2012*). NIS Elements software was used to subtract background and measure fluorescence intensity in pHluorin and mCherry channels. For larval measurements, fluorescence intensity measurements were collected from 3 ROI per eye disc,

with each ROI comprising ~7 ommatidia. For pupal measurements, fluorescence intensity values in single cells were measured for 10 cells per type per eye; cell types were identified by their stereotyped morphology at either the apical surface (cone cells, primary, secondary and tertiary pigment cells) or in the middle of the retinal epithelium (photoreceptor neurons). Ratiometric calculations and statistical analyses were performed in Microsoft Excel. To generate a standard curve for each genotype, tissue was incubated for 20–40 min in nigericin buffer (25 mM HEPES, 105 mM KCl, 1 mM MgCl$_2$, 20 μM nigericin [Invitrogen]) at defined pH points (~6.5, ~7.0 and ~7.5). We did observe a difference in pHi in wild type between our two sets of experiments shown in *Figure 1* and *Figure 3*, which we attribute to alterations in our imaging systems made between the sets of experiments. Importantly, significant differences were reproducibly seen between genotypes in both series of studies.

## Cell culture

Estrogen-inducible H-RasV12 (ER-RasV12) cells were a gift from Z Gartner. Control, stable H-RasV12-expressing (sRasV12) and vector control MCF10A cells were a gift from J Debnath. Cell lines were authenticated by DNA sequencing (DDC Medical, Fairfield, OH). Cells were cultured as described (*Debnath et al., 2003*). SW-620 and MDA-MB-231 cells were obtained from ATCC (Manassas, VA) and cultured in DMEM-H21 high glucose media + 10% FBS at 37°C with 5% CO$_2$; cells were serum-starved (0.2% FBS) for experiments. NHE1 activity was determined as the rate of pHi recovery from an NH$_4$Cl-induced acid load in a nominally HCO$_3^-$-free buffer as described (*Denker et al., 2000*). Cells were treated with the NHE1 inhibitor 5(N-ethyl-N-isopropyl) amiloride (EIPA) at 10 μM for 48 or 72 hr prior to assay. For NHE1 shRNA treatment, lentiviral shRNA constructs (TRCN0000044650 and TRCN0000044648) were purchased from Sigma–Aldrich (Saint Louis, MO).

After 2 days of lentiviral infection, cells were selected in 5 μg/ml puromycin for 72 hr and acid load selection was performed as described (*Pouysségur et al., 1984*) and plated for experiments within 2 weeks of initial lentiviral infection. pHi measurements were performed as described (*Denker et al., 2000*). FITC-labeled AnnexinV was performed according to manufacturer's directions (Becton, Dickinson and Company, Franklin Lakes, NJ, cat #560931) on a BD Accuri C6 Flow Cytometer. Trypan Blue labeling was performed according to manufacturer's directions (Life Technologies) and cells were counted using a hemocytometer.

## Acknowledgements

We thank Torsten Wittmann, Lutz Kockel, and members of the Barber, Wittmann and Katja Brückner labs for helpful discussions and Andrei Goga, Tom Kornberg, Sarah Knox, and Todd Nystul for helpful comments on the manuscript. We also thank Ross Cagan and Dave Casso for sharing *Drosophila* lines, Zev Gartner and Jay Debnath for providing MCF10A cells expressing oncogenic Ras, Peter Sargent for use of instrumentation for histological sample preparation, Richard Jordan and Scott Oakes for assistance with histological analysis, Larry Ackerman for assistance preparing SEM samples, and members of the Scott Oakes and Max Krummel labs for training in FACS analysis. This work was supported by NIH R01 GM58642, a UCSF Academic Senate grant, and a University of California Cancer Research Coordinating Committee Award.

## Additional information

### Funding

| Funder | Grant reference | Author |
|---|---|---|
| National Institutes of Health | R01 GM58642 | Diane L Barber |
| University of California, San Francisco | Cancer Research Coordinating Committee Award | Diane L Barber |
| University of California, San Francisco | Academic Senate grant | Diane L Barber |

The funders had no role in study design, data collection and interpretation, or the decision to submit the work for publication.

## Author contributions

BKG-H, DLB, Conception and design, Acquisition of data, Analysis and interpretation of data, Drafting or revising the article; CC, Acquisition of data, Analysis and interpretation of data; MJ-V, Generated *Dnhe2^null^ Drosophila* lines, Contributed unpublished essential data or reagents

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
