## [Decision Letter]

Thank you for sending your work entitled “Increased H^+^ Efflux is Sufficient to Induce Dysplasia and Necessary for Viability with Oncogene Expression” for consideration at *eLife*. Your article has been evaluated by Charles Sawyers (Senior editor) and four reviewers, one of whom is a member of our Board of Reviewing Editors.

The reviewers agree that the research is interesting and potentially very exciting, putting intracellular pH as a primary contributor to carcinogenesis. However, the reviewers also agree that the phenotype is not well defined, and proper characterization will require additional experiments, probably including the analysis of new fly crosses. The reviewers feel that the work may well take more than the 2 month revision time frame, because new strains will likely be needed and this will be time consuming. You will need to decide whether to revise for *eLife* or withdraw and try another journal.

The reviewers have discussed their major comments and arrived at the following consensus comments that will need to be addressed in a revised submission.

1) More analysis is needed to define the phenotypes of Nhe2 overexpression and loss of function. The present analysis is inadequate to determine whether the effects are consistent with altered proliferation, migration or other biology, as suggested by in vitro studies of NHE1. The standard approach for analysis of *Drosophila* oncogenes and tumor suppressors is to compare clones of cells that do or do not contain the genetic alteration, typically using the FRT/Flipout system to generate mutant clones and twin spots. You should make clones of *Dnhe2* mutant cells in the proliferating epithelium (the eye or wing disc) and compare them to the twin-spot cloned to see whether the mutant clone has a change in cell number or cell size, and whether the labeled mutant cells have migrated away from their original location or invaded to other tissues. This will circumvent the issues of high lethality in some of your experiments.

2) The use of the GMR driver for expression means that Nhe2 is being over expressed in differentiating photoreceptors and accessory cells. This may compromise your ability to detect an effect on cell proliferation. A driver expressed in proliferating cells to allow you to detect changes in proliferation (e.g. additional cycles after the normal exit time).

3) A pHi indicator driven from an ubiquitous promoter should be used to allow comparison of pHi in altered clones with surrounding normal cells.

If these experiments allow clear conclusions as to the effects of Nhe2 gain and loss of function in conjunction with Ras and Raf oncogenes, then some of the more paradoxical results (e.g., reduced/increased pHi upon Nhe2 loss in larva/pupa) could be omitted and reserved for a subsequent paper in which the mechanisms are investigated.

---

## [Author Response]

*1) More analysis is needed to define the phenotypes of Nhe2 overexpression and loss of function. The present analysis is inadequate to determine whether the effects are consistent with altered proliferation, migration or other biology, as suggested by in vitro studies of NHE1. The standard approach for analysis of* Drosophila *oncogenes and tumor suppressors is to compare clones of cells that do or do not contain the genetic alteration, typically using the FRT/Flipout system to generate mutant clones and twin spots. You should make clones of* Dnhe2 *mutant cells in the proliferating epithelium (the eye or wing disc) and compare them to the twin-spot cloned to see whether the mutant clone has a change in cell number or cell size, and whether the labeled mutant cells have migrated away from their original location or invaded to other tissues. This will circumvent the issues of high lethality in some of your experiments*.

As requested, our revision includes new data and additional text for a more thorough phenotypic analysis of *Dnhe2* overexpression and loss-of-function. New data on the *GMR>Dnhe2* phenotype in pupal retinae show clear effects on cell fate specification, cell-cell contacts and cell morphologies across cell types (Figure 2). We also include new data from generating mosaic tissue to characterize *Dnhe2* overexpression phenotypes in wing epithelium. We expressed *Dnhe2* in the posterior compartment of the developing wing imaginal disc using an *engrailed* driver (*enGAL4*). We show autonomous effects of *Dnhe2* on proliferation and on dysplasia (Figure 2). We do not see evidence of *en>Dnhe2*-expressing cells migrating out of the tissue across compartment boundaries, similar to our results using *ptcGAL4* (Figure 3), suggesting that *Dnhe2* is not sufficient to induce migration alone.

We also include new data on *Dnhe2* enhancing oncogene-induced invasion. To determine effects on invasive phenotypes, we identified invasive cells based on their isolation in three dimensions from other GFP-positive cells in the wing disc. We found only a single example where a few (56; 48; 41) cells that met this definition in either control or *Dnhe2*-expressing discs, and no examples in *RasV12*-expressing discs (Figure 2). However, in *ptc>RasV12, Dnhe2* discs, we found examples of invasive cells in all discs examined. Further, we found several different types of invasive behaviors, including basal expansion of the *ptc* stripe, single invasive cells and streams of cells migrating out of the stripe into neighboring tissue (Figure 2).

Finally, we added new data on cell death with inhibiting NHE1 in two more tumor-derived cell types, MDA-MB-231 clonal mammary carcinoma cells and SW620 clonal colorectal cells.

We attempted but were unsuccessful in using twinspot analysis with *Dnhe2* over-expression. We spent several months generating the *Drosophila* lines necessary for determining whether over-expression of *Dnhe2* increased clone size relative to a twinspot. We then spent two months testing heat-shock conditions (altered temperature and length of heat shock) for generating 1-2 clones per disc, which is necessary for analysis. However, we either obtained no clones or we obtained too many clones to properly match clone and twinspot.

We are unable to generate loss-of-function mosaic clones because the *Dnhe2* genomic locus is too close to the nearest flippase recognition site (FRT40A).

*2) The use of the GMR driver for expression means that Nhe2 is being over expressed in differentiating photoreceptors and accessory cells. This may compromise your ability to detect an effect on cell proliferation. A driver expressed in proliferating cells to allow you to detect changes in proliferation (e.g. additional cycles after the normal exit time)*.

As requested, we performed additional experiments to test the effects of overexpressing *Dnhe2* using a different driver. We used two transgenic lines that contain the *eyeless-GAL4* (*eyGAL4*) driver, which is expressed in retinal cells from their specification during embryogenesis and crossed them to two different *UAS-Dnhe2* transgene insertion lines. We found that eyes in *ey>Dnhe2* lines showed a less disrupted architecture than *GMR>Dnhe2*, which may be because *GMRGAL4* is a very strong driver and likely generates higher *Dnhe2* expression than *eyGAL4*.

*3) A pHi indicator driven from an ubiquitous promoter should be used to allow comparison of pHi in altered clones with surrounding normal cells*.

This is an excellent suggestion, and we agree this would be an ideal strategy to compare pHi in altered clones with surrounding normal cells. However, the suggested strategy is currently not technically feasible because it would require three fluorophores with non-overlapping spectra. In our transgenic lines, the pH-sensitive pHluorin (ex475, em535) is fused to pH-insensitive mCherry (ex587, em610) to normalize to expression of the biosensor. Therefore, only blue or far-red fluorophores could be used to mark clones. CFP (ex435, em485) cannot be used because it has spectral overlap with GFP. Existing fluorescent proteins toward the far-red end of the spectrum such as mKate (ex588, em635), E2-Crimson (ex611, em646) or mPlum (ex590, em649) have spectral overlap with mCherry.